# How Quickly Do Proteins Fold and Unfold, and What Structural Parameters Correlate with These Values?

**DOI:** 10.3390/biom10020197

**Published:** 2020-01-29

**Authors:** Anna V. Glyakina, Oxana V. Galzitskaya

**Affiliations:** 1Institute of Protein Research, Russian Academy of Sciences, Pushchino, Moscow Region 142290, Russia; glyakina@rambler.ru; 2Institute of Mathematical Problems of Biology, Russian Academy of Sciences, Keldysh Institute of Applied Mathematics, Russian Academy of Sciences, Pushchino, Moscow Region 142290, Russia; 3Institute of Theoretical and Experimental Biophysics, Russian Academy of Sciences, Pushchino, Moscow Region 142290, Russia

**Keywords:** protein unfolding, unfolding rates, bacterial proteins, eukaryotic proteins, thermophile, mesophile

## Abstract

The correlations between the logarithm of the unfolding rate of 108 proteins and their structural parameters were calculated. We showed that there is a good correlation between the logarithm of folding and unfolding rates (0.79) and protein stability and unfolding rate (0.79). Thus, the faster the protein folds, the faster it unfolds. Folding and unfolding rates are higher for the proteins with two-state kinetics, in comparison with the proteins with multi-state kinetics. At the same time, two-state bacterial proteins folds and unfolds two orders of magnitude faster than two-state eukaryotic proteins, and multi-state bacterial proteins folds and unfolds slower than multi-state eukaryotic proteins. Despite the fact that the folding rates of thermophilic and mesophilic proteins are close, the unfolding rates of thermophilic proteins is about two orders of magnitude lower than for mesophilic proteins. The correlation between unfolding rate and stability of thermophilic proteins is high (0.90). We also found that the unfolding rate correlates with such structural parameters as: size of the protein, radius of the cross-section, logarithm of absolute contact order, and radius of gyration. This information will be useful for engineering and designing new proteins with desired properties.

## 1. Introduction

The problem of predicting folding rates (*kf*) for proteins with two-state and multi-state kinetics is still important and extensively studied [1,2,3,4,5,6,7,8,9,10,11,12,13,14,15,16,17,18,19,20]. Many articles are devoted to the study of protein folding rates and their correlation with various structural parameters [2,6,7,8,16]. In 1998, a relative contact order (rCO) parameter was suggested, which is the average distance along the sequence between all pairs of contacting residues, normalized to the size of the protein (number of amino acid residues, further protein length). This parameter reflects the topological complexity of the protein chain. It was shown that the rCO correlates well (correlation coefficient is 0.81) with the logarithm of the folding rate for 12 two-state proteins [2]. Subsequent studies have shown that there is no correlation between rCO and logarithm of the folding rate of proteins [6,7,16]. It turned out that only absolute contact order (AbsCO, contact order multiplied by protein length) correlated with the logarithm of the folding rate (the correlation coefficient is −0.77) [16]. It was found that the structural parameters, depending on the protein length (L), correlated well with the logarithm of the folding rate [16]. In the set of papers [13,15,16,21,22], the authors considered the different structural parameters of protein globule compactness: radius of gyration (R_g_); normalized radius of gyration (R_g_/R_g_*, where R_g_* is the radius of gyration of the ball with the uniform density and the same volume); compactness as the relationship between the solvent accessible surface area to the solvent accessible surface of sphere with the same volume (S_asa_/S*); and the relationship between the solvent accessible volume to the solvent accessible surface (V_asa_/S_asa_). The solvent accessible volume (V_asa_) is the volume surrounded by the solvent accessible surface (S_asa_). The radius of the cross-section (V_asa_/S_asa_) is proportional but not equal to the radius of the smallest cross-section of the protein globule [16]. It was demonstrated that V_asa_/S_asa_ correlates good with the logarithm of the folding rate (the correlation coefficient is 0.74) [16]. We have shown that the less compact the proteins, the faster they fold. Therefore, α-structural proteins are less compact and the most rapidly folding proteins, while α/β-structural proteins are the most compact and the most slowly folding proteins [15].

We have previously shown that bacterial proteins with two-state kinetics (further, bacterial two-state proteins) fold and unfold faster than two-state eukaryotic proteins [23]. It turned out that the multi-state eukaryotic proteins fold and unfold faster than bacterial ones.

It has been shown that there is a “golden triangle” that limits the possible range of the folding rates for single-domain globular proteins of different sizes and stability [24]. This triangle was based only on the biological and physical limitations.

In addition, knowledge of the protein unfolding rates is also necessary to estimate their stability. Much less work has been devoted to the study of the relationship between the unfolding rates and structural parameters. It has been shown that the logarithm of the unfolding and folding rates correlate strongly with the AbsCO and the long-range order (parameter defined by contacts between two residues that are close in space and far in the sequence) [3], with correlation coefficient values of 0.75 or higher. The logarithm of the unfolding rate correlated better with thermodynamic stability, in comparison with the folding rate [19].

Early, it has been shown that structural parameters L, ln(AbsCO), V_asa_/S_asa_, and R_g_ were in a good correlation with the logarithm of the protein folding rate [16].

The aim of the study was broader than the search for connections between protein unfolding rates and various structural parameters, such as L, ln(AbsCO), V_asa_/S_asa_, and R_g_. We were trying to find out which parameters are most important for the prediction of the unfolding rates for proteins from different organisms.

## 2. Materials and Methods 

In our study, three databases of proteins were examined. The first database consisted of 108 proteins. Among them, 38 proteins demonstrated multi-state kinetics, and 70 proteins showed two-state kinetics. The second database consisted of 42 bacterial (29 two-state and 13 multi-state) and 53 eukaryotic (32 two-state and 21 multi-state) proteins. The third database consisted of 42 bacterial proteins. Among them, there were 10 thermophilic (8 two-state and 2 multi-state) and 32 mesophilic (21 two-state and 11 multi-state) proteins.

Proteins that have experimentally measured folding and unfolding rates were taken into consideration. The database of such proteins has begun to be collected since 2003. At that time, there were only 57 proteins [7]. In 2009, there were already 84 proteins [16]. Now, this database consists of 108 proteins [19,24].

Data on folding and unfolding rates and structural parameters of the proteins (L—length of the protein, V_asa_/S_asa_, ln(AbsCO) and R_g_) are in Appendix A. 

Parameters V_asa_ and S_asa_ were calculated using YASARA program [25]. 

Parameter ln(AbsCO) was calculated as $ln\left(AbsCO\right)=ln(\frac{1}{N}\overset{N}{\underset{k=1}{\sum }}\Delta L_{ij})$, where *N* is the number of contacts (within 6 Å) between nonhydrogen atoms in the protein and *∆L_ij_* is the number of residues separating the interacting pair of nonhydrogen atoms (adjacent residues are assumed to be separated by one residue).

Parameter R_g_ was calculated as $R_{g}=\sqrt{\frac{\sum^{\;}m_{i}{\left(r_{i}-R_{C}\right)}^{2}}{M}}$, where *m_i_* is the mass of the *i*-th atom, *r_i_* is its Cartesian coordinates, *M* is the mass of the protein, and *R_C_* is the coordinate vector of the mass center of the protein, calculated as follows: $\sum^{\;}m_{i}\left(r_{i}-R_{C}\right)=0$.

Errors for data from Table 1, Table 3, Table 5, Figure 5 and Figure 9 were calculated as $\sigma =\sqrt{\frac{\sum_{i=1}^{n}\left(xi-\bar{x}\right)}{n\left(n-1\right)}}$, where $\bar{x}$ is the average value of a parameter and *n* is a number of proteins.

## 3. Results and Discussion

### 3.1. Unfolding Rates of 108 Proteins

The goal of this work was to find the relationship between protein unfolding rates and protein structural parameters. For three spectrin domains: R15, R16, and R17, we observed that the faster the protein folds, the faster it unfolds, and vice versa. Domain R15 folds and unfolds faster than its homologues, R16 and R17 (see Figure 1A). In the case of these spectrin repeats, the folding and unfolding rates may be associated with the mechanical stability of the proteins. Previously, it has been shown that domain R15 is less mechanically stable than domains R16 and R17 [26]. The discovered correlation between the unfolding and folding rates suggests that the statement that the faster the protein folds, the faster it unfolds, and vice versa, is also true for a dataset consisting of 108 proteins (Figure 1B). 

The correlation between the logarithm of the unfolding and folding rates is 0.79 for all proteins. Moreover, this correlation is better for two-state (0.78) than for multi-state proteins (0.73). The separation of 108 proteins by structural classes (α, β, α/β and α + β) revealed that correlation between the logarithm of the folding and unfolding rates is better for proteins from α and β (0.78 and 0.75) classes, in comparison with the proteins from α/β and α + β classes (0.59 and 0.60). Moreover, two-state proteins make the largest contribution to this correlation (see Table 1). 

The proteins from α class folds and unfolds faster, while proteins from α/β class folds and unfolds slowly, in comparison with proteins from other structural classes (average logarithm of folding rates: 8.49 ± 0.64 for α, 3.42 ± 0.63 for β, −0.02 ± 0.85 for α/β, and 4.71 ± 0.53 for α + β; average logarithm of unfolding rates: 2.03 ± 1.03 for α, −4.51 ± 1.12 for β, −8.34 ± 1.64 for α/β, and −4.76 ± 0.97 for α + β; see also Appendix A).

It was previously shown that L, ln(AbsCO), V_asa_/S_asa_, and R_g_ correlate well with the logarithm of the protein folding rate [16]. Thus, it can be assumed that if these parameters correlate well with the logarithm of the folding rate, then they will also correlate well with the logarithm of the unfolding rate. In this case, four parameters were examined: L is a number of amino acid residues in protein, ln(AbsCO) is the logarithm of the absolute contact order, V_asa_/S_asa_ is a radius of cross-section, and R_g_ is a radius of gyration.

The values of structural parameters considered in this paper (L, ln(AbsCO), V_asa_/S_asa_ and R_g_) are lower for two-state proteins than for multi-state proteins: 78 ± 5 vs. 130 ± 8 for L, 3.14 ± 0.05 vs. 3.59 ± 0.06 for V_asa_/S_asa_, 6.91 ± 0.06 vs. 7.22 ± 0.06 for ln(AbsCO), and 12.1 ± 0.3 vs. 14.2 ± 0.3 for R_g_ (see Table 2). The logarithms of the folding and unfolding rates are higher for two-state proteins, in comparison with multi-state proteins: 6.08 ± 0.50 vs. 2.51 ± 0.59 for the folding rate and −1.51 ± 0.79 vs. −6.09 ± 1.03 for the unfolding rate, respectively (see Figure 1B). 

For 108 proteins, the correlations between the logarithm of the unfolding rate (ln(*ku*)) and structural parameters such as L, V_asa_/S_asa_, ln(AbsCO), and R_g_ were calculated (Table 3 and Figure 2). V_asa_/S_asa_ and ln(AbsCO) are better correlated with the logarithm of the unfolding rate of two-state proteins. For two-state proteins, these correlations are −0.79 and −0.87, in comparison with −0.63 and −0.69 for multi-state proteins. The correlation between R_g_ and the logarithm of the unfolding rate is almost the same for two-state and multi-state proteins (−0.61 vs. −0.60, respectively). Moreover, L is better correlated with the logarithm of the unfolding rate of multi-state proteins. Good correlation (0.79) between the protein stability (−(ln*kf* − ln*ku*)) and the logarithm of the unfolding rate has been observed.

After the separation of 108 proteins by structural classes (α, β, α/β and α + β), we observed that correlations between the logarithm of the unfolding rate (*ln(ku)*) and L, V_asa_/S_asa_, ln(AbsCO), and R_g_ are better for proteins from α and β classes (see Table 4). These correlations are the highest for proteins from β class (higher than 0.8). The largest contribution to these correlations made two-state proteins (see Table 4). The exception is only for correlation between *ln(ku)* and L for proteins from β class. This correlation is higher for multi-state proteins (−0.86), in comparison with two-state proteins (−0.84). 

### 3.2. Unfolding Rates of Bacterial and Eukaryotic Proteins

To find the dependence of the unfolding rates on the origin of the proteins, the 42 bacterial and 53 eukaryotic proteins from our database were separately studied. Two-state bacterial proteins fold and unfold faster than two-state eukaryotic proteins. For multi-state proteins, we observed that bacterial proteins fold and unfold slower than eukaryotic proteins (see Figure 3 and Figure 4). The same result was observed when the dataset consisted of 35 bacterial and 38 eukaryotic proteins [23]. The correlation between the logarithm of the unfolding and folding rates is 0.73 for bacterial and 0.75 for eukaryotic proteins. Moreover, for bacterial proteins, this correlation is better for two-state (0.69) than for multi-state proteins (0.45). For eukaryotic proteins, this correlation is better for multi-state (0.81) than for two-state proteins (0.72). Values V_asa_/S_asa_, ln(AbsCO), and R_g_ are slightly higher for the bacterial proteins, and this gap increases for multi-state proteins: 3.74 ± 0.07 vs. 3.47 ± 0.07 for V_asa_/S_asa_, 7.40 ± 0.07 vs. 7.14 ± 0.10 for ln(AbsCO), and 14.8 ± 0.5 vs. 13.9 ± 0.4 for R_g_, respectively (Table 5).

Then, the correlations between the logarithm of the unfolding rate and structural parameters for bacterial and eukaryotic proteins were investigated. The correlations between the logarithm of the unfolding rate and L, V_asa_/S_asa_, and ln(AbsCO) are almost the same for all bacterial and eukaryotic proteins: −0.67 vs. −0.68 for L, −0.72 vs. −0.69 for V_asa_/S_asa_, and −0.80 vs. −0.79 for ln(AbsCO), respectively (Table 6 and Figure 4). The difference is observed only for R_g_, which correlates better with the logarithm of the unfolding rate of bacterial proteins (−0.71). If we consider these correlations for two-state and multi-state bacterial and eukaryotic proteins separately, we get the following picture. For two-state proteins, the correlation between the logarithm of the unfolding rate and V_asa_/S_asa_ is almost the same for bacterial and eukaryotic proteins (−0.75 vs. −0.77). R_g_ and ln(AbsCO) better correlate with the logarithm of the unfolding rate of two-state bacterial proteins than with eukaryotic proteins (−0.86 vs. −0.77 for ln(AbsCO) and −0.64 vs. −0.47 for R_g_, respectively). L, on the contrary, correlates better with the logarithm of the unfolding rate of two-state eukaryotic proteins (−0.61 vs. −0.75). For multi-state proteins, we observed the same picture as for two-state proteins for correlations of L and R_g_ with the logarithm of the unfolding rate. Both V_asa_/S_asa_ and ln(AbsCO) correlate better with the logarithm of the unfolding rate of multi-state eukaryotic proteins than with bacterial proteins: −0.11 vs. −0.69 for V_asa_/S_asa_ and −0.22 vs. −0.81 for ln(AbsCO), respectively.

Amino acid composition of bacterial and eukaryotic proteins was analyzed (Figure 5). The bacterial proteins with two-state kinetics are enriched in Ala, Gly, Lys, and Asn, compared with eukaryotic proteins with two-state kinetics. The eukaryotic proteins with two-state kinetics contain more His, Leu, Pro, Arg, Ser, and Trp, compared to the bacterial proteins with two-state kinetics (see Figure 5).

### 3.3. Unfolding Rates of Proteins from Thermophilic and Mesophilic Organisms

Since a lot of attention was paid to the search for differences between thermophilic and mesophilic proteins — in particular, folding rates — we also decided to conduct our analysis for these proteins. All bacterial proteins were divided into thermophilic and mesophilic groups. Further in the text, we call proteins from thermophilic organisms as thermophilic proteins and proteins from mesophilic organisms as mesophilic proteins. The correlation between the logarithm of the unfolding and folding rates is better for mesophilic (0.76), in comparison with thermophilic (0.73) proteins. Moreover, for mesophilic proteins, this correlation is better for two-state (0.76) than for multi-state proteins (0.12). For thermophilic proteins, it is hard to say something, because there are only two proteins with multi-state kinetics. There is a correlation between stability and the logarithm of the unfolding rate for thermophilic (0.90) and mesophilic (0.73) proteins. The logarithm of the folding rate of thermophilic and mesophilic proteins are almost the same (4.75 ± 1.20 vs. 4.58 ± 0.79) (Figure 6 and Table 7). Still, mesophilic proteins unfold faster than thermophilic proteins (−5.63 ± 2.31 vs. −3.27 ± 1.12). The same picture is observed for two-state thermophilic and mesophilic proteins. Schematic “chevron” plots for thermophilic and mesophilic proteins are presented in Figure 7.

Finally, the correlations of the logarithm of the unfolding rate and structural parameters for thermophilic and mesophilic proteins were examined (Table 8 and Figure 8). L and R_g_ correlate better with the logarithm of the unfolding rate of all thermophilic proteins (−0.83 vs. −0.64 for L and −0.87 vs. −0.66 for R_g_), and ln(AbsCO) correlates better with the logarithm of the unfolding rate of all mesophilic proteins (−0.74 vs. −0.83). V_asa_/S_asa_ correlates with the logarithm of the unfolding rate practically the same for all thermophilic and mesophilic proteins (−0.77 vs. −0.75). For two-state thermophilic proteins, the correlation between all considered parameters (L, V_asa_/S_asa_, ln(AbsCO) and R_g_) is better than for two-state mesophilic proteins: −0.92 vs. −0.60 for L, −0.92 vs. −0.78 for V_asa_/S_asa_, −0.93 vs. −0.85 for ln(AbsCO), and −0.88 vs. −0.60 for R_g_.

The thermophilic proteins are enriched with Lys, Arg, and Val, in comparison with the mesophilic proteins, and enriched in Lys, Asp, Ala, and the mesophilic proteins contain more Asp, Asn, Ser, and Thr, in comparison with the thermophilic proteins (Figure 9). The same can be said about two-state thermophilic and mesophilic proteins. These data are also consistent with those that we obtained earlier in the study of 373 pairs of structurally similar thermophilic and mesophilic proteins [27].

## 4. Discussion

In this paper, we tried to find parameters that are important for predicting the protein unfolding rates. For this, the database consists of 108 proteins with known unfolding and folding rates, and such structural parameters as L, ln(AbsCO), V_asa_/S_asa_, and R_g_ were considered.

The good correlation (0.79) between the logarithm of the unfolding rate and protein stability was observed for 108 proteins.

First, we divided the proteins in our database into two-states and multi-states. On average, the logarithms of the folding and unfolding rates are higher for two-state proteins, in comparison with multi-state proteins. A good correlation (not lower than 0.70) for the logarithm of the folding and unfolding rates for two- and multi-state proteins was observed. It has been shown that the logarithm of the unfolding rate of two-state proteins correlate better with V_asa_/S_asa_ (−0.79) and ln(AbsCO) (−0.87), and the logarithm of the unfolding rate of multi-state correlates better with L (−0.71).

Then, we separately studied bacterial and eukaryotic proteins from our database. It has been shown that two-state bacterial proteins fold and unfold faster than two-state eukaryotic proteins, and multi-state eukaryotic proteins fold and unfold faster than multi-state bacterial proteins. The logarithm of the unfolding rate of two-state bacterial proteins correlates better with ln(AbsCO) (−0.86) and R_g_ (−0.64), and eukaryotic proteins correlate better with L (−0.75). For multi-state proteins, the following picture is observed: the logarithm of the unfolding rate of bacterial proteins correlates better with R_g_ (−0.58), and eukaryotic proteins correlate better with L (−0.77), V_asa_/S_asa_ (−0.69), and ln(AbsCO) (−0.81).

Finally, we separately studied the thermophilic and mesophilic bacterial proteins from our database. There is correlation of the logarithm of the unfolding rate with protein stability for thermophilic (0.90) and mesophilic proteins (0.73). It has been shown that the logarithm of the unfolding rates of thermophilic proteins are about two orders of magnitude lower than that of mesophilic proteins, but the logarithm of the folding rates of thermophilic and mesophilic proteins are almost the same. The logarithm of the unfolding rate of two- and multi-state thermophilic proteins correlate better with all considered structural parameters (L, V_asa_/S_asa_, ln(AbsCO) and R_g_), in comparison with the mesophilic proteins.

We have tried to find out which parameters are most important for the prediction of the unfolding rates for proteins from different structural classes (α, β, α/β and α + β); proteins of different origins (bacterial and eukaryotic); and proteins from different organisms (thermophilic and mesophilic).

## 5. Conclusions

Thus, it has been shown that there is a good correlation between the logarithm of the unfolding and folding rates (0.79) and between the logarithm of the unfolding rate and proteins stabilities (0.79) for 108 proteins. The correlation between the unfolding and folding rates is better for: two-state (0.78), in comparison with multi-state (0.73) proteins; α and β proteins (0.78 and 0.75), in comparison with α/β and α + β protein (0.59 and 0.60) structural classes; eukaryotic (0.75), in comparison with bacterial (0.73) proteins; and mesophilic (0.76), in comparison with thermophilic (0.73) proteins. The structural parameter ln(AbsCO) better correlates with the logarithm of the unfolding rate for: all 108 proteins; proteins from α and β structural classes; and bacterial, eukaryotic, and mesophilic proteins, in comparison with other parameters (L, V_asa_/S_asa_ and R_g_).

## Figures and Tables

**Figure 1 biomolecules-10-00197-f001:**
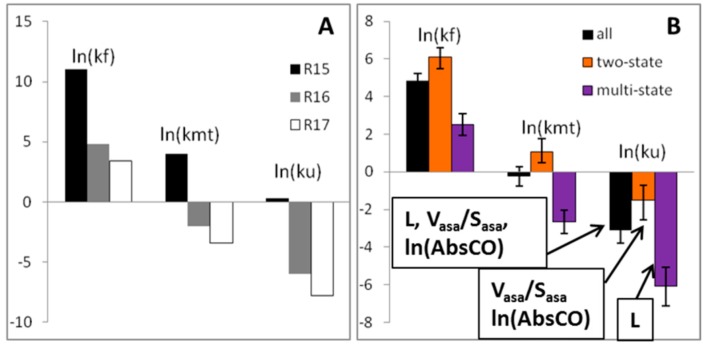
(**A**) Logarithm of the unfolding and folding rates for the three domains of chicken brain alpha-spectrin proteins. (**B**) The average values of the logarithm of the unfolding and folding rates for 108 proteins. Structural parameters that better correlate with the logarithm of the unfolding rates are outlined in a rectangle.

**Figure 2 biomolecules-10-00197-f002:**
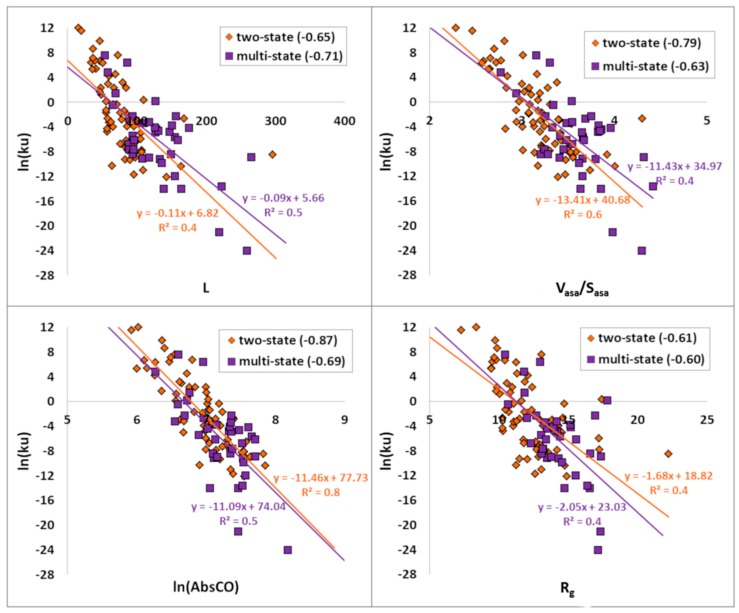
Correlations of the logarithm of the unfolding rates of 108 proteins with their structural parameters: L—length of the protein, V_asa_/S_asa_—radius of the cross-section, ln(AbsCO)— logarithm of the absolute contact order, and R_g_ (radius of gyration). There is a line approximation of points and its equation: orange line corresponds to two-state proteins and purple line to multi-state proteins. R^2^ is a linear approximation reliability.

**Figure 3 biomolecules-10-00197-f003:**
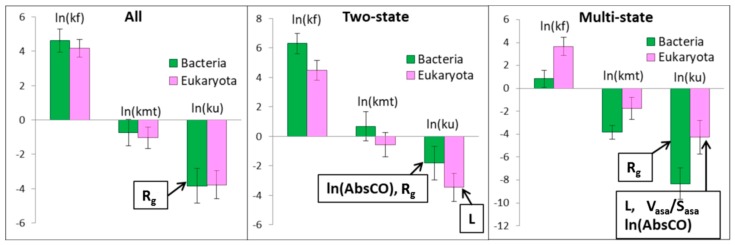
The average values of the logarithm of the unfolding and folding rates for bacterial and eukaryotic two- and multi-state proteins. Structural parameters that better correlate with the logarithm of the unfolding rate are outlined in a rectangle.

**Figure 4 biomolecules-10-00197-f004:**
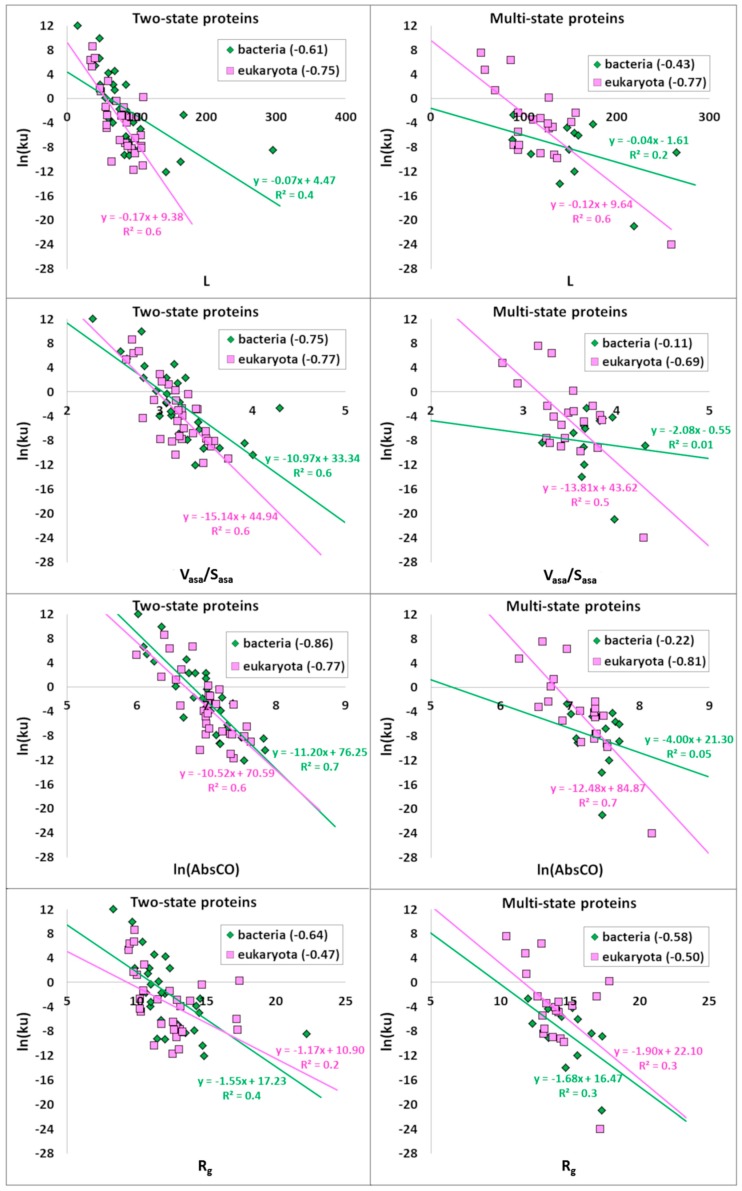
Correlations of the logarithm of the unfolding rate of the two- and multi-state bacterial and eukaryotic proteins with structural parameters: L, V_asa_/S_asa_, ln(AbsCO), and R_g_. There is a line approximation of points and its equation: green line corresponds to bacterial proteins and pink line to eukaryotic proteins. R^2^ is a linear approximation reliability.

**Figure 5 biomolecules-10-00197-f005:**
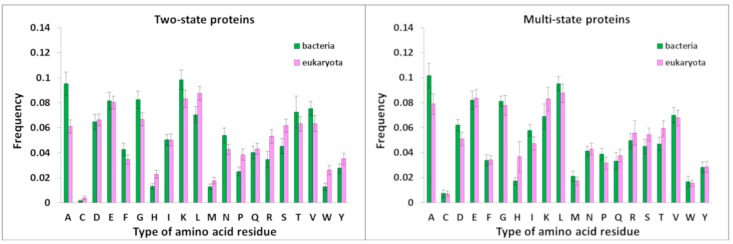
Amino acid composition of the bacterial and eukaryotic proteins with two- and multi-state kinetics.

**Figure 6 biomolecules-10-00197-f006:**
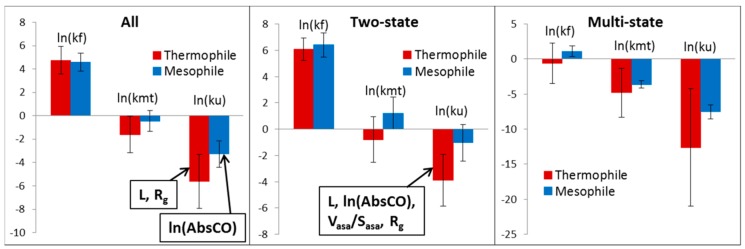
The average values of the logarithm of the unfolding and folding rates for thermophilic and mesophilic two- and multi-state proteins. Structural parameters that better correlate with the logarithm of the unfolding rate are outlined in a rectangle.

**Figure 7 biomolecules-10-00197-f007:**
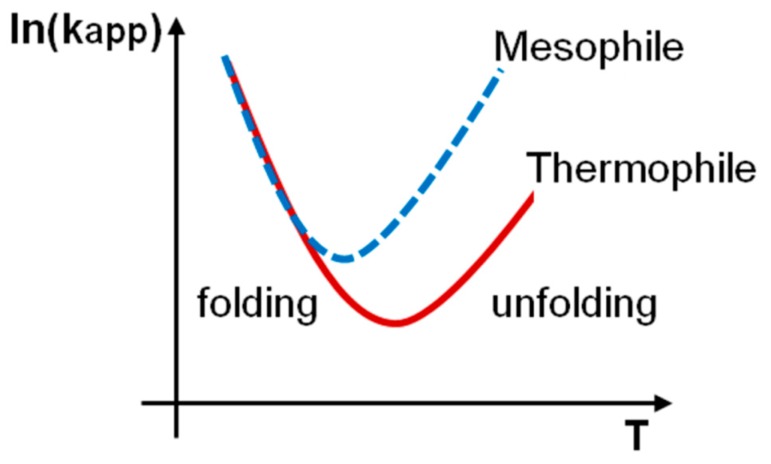
Schematic “chevron” plots of the constant of observed rate (*kapp = kf + ku*) versus temperature for thermophilic and mesophilic proteins.

**Figure 8 biomolecules-10-00197-f008:**
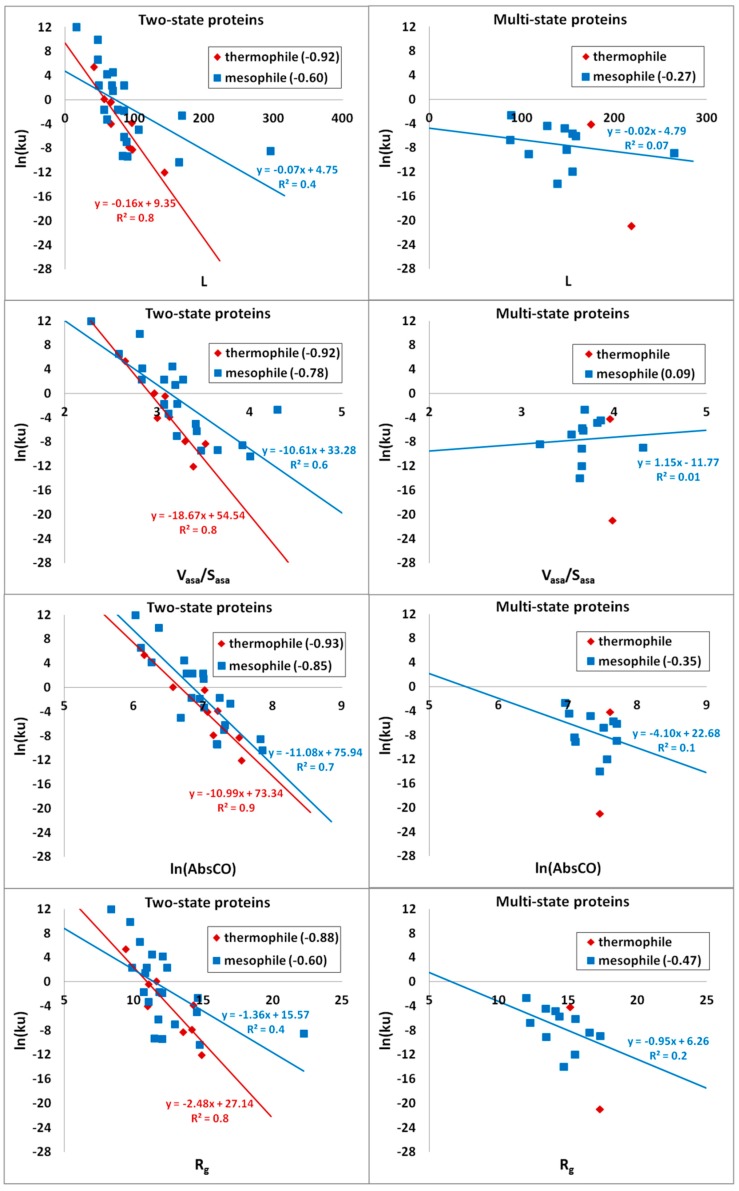
Correlations of the logarithm of the unfolding rate of the two- and multi-state thermophilic and mesophilic proteins with structural parameters: L (length of the protein), V_asa_/S_asa_, ln(AbsCO), and R_g_. There is a line approximation of points and its equation: red line corresponds to thermophilic proteins and blue line to mesophilic proteins. R^2^ is a linear approximation reliability.

**Figure 9 biomolecules-10-00197-f009:**
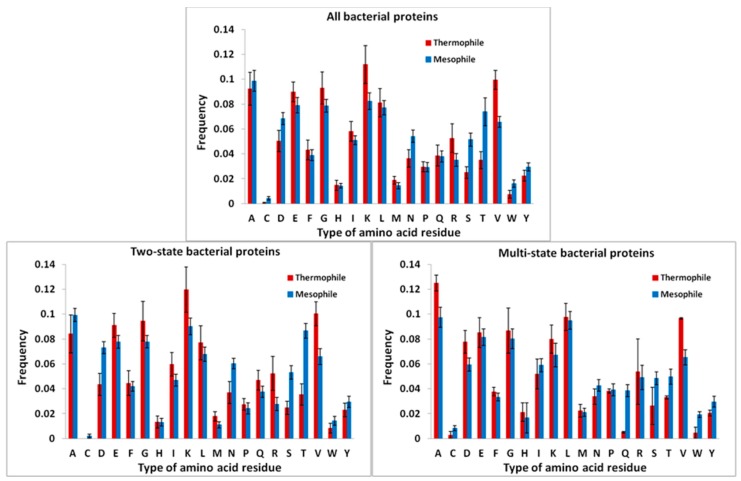
Amino acid composition of the bacterial thermophilic and mesophilic proteins with two- and multi-state kinetics.

**Table 1 biomolecules-10-00197-t001:** Correlations between logarithm of unfolding (ln(*ku*) and folding (ln(*kf*) rates for four structural classes of proteins α, β, α/β and α + β.

Correlations with ln(*ku*)	All (Two + Multi)	Two-State	Multi-State
All	**0.79*** (108) **	**0.78** (70) **	0.73 (38) **
α	**0.78** (31)	**0.82** (23)	**0.79** (8)
β	**0.75** (37)	0.74 (25)	0.61 (12)
α/β	0.59 (12)	-	0.55 (11)
α + β	0.60 (28)	0.63 (21)	0.13 (7)

* Correlations above 0.75 are shown in bold. ** The number of proteins is indicated in parentheses.

**Table 2 biomolecules-10-00197-t002:** Average values of structural parameters for 108 proteins. V_asa_/S_asa_ = radius of cross-section. R_g_ = radius of gyration. ln(AbsCO) = logarithm of the absolute contact order. L = length of the protein.

Average Value	All (Two + Multi)	Two-State	Multi-State
Number of proteins	108	70	38
<L>	96 ± 5	78 ± 5	130 ± 8
<V_asa_/S_asa_>	3.30 ± 0.04	3.14 ± 0.05	3.59 ± 0.06
<ln(AbsCO)>	7.02 ± 0.05	6.91 ± 0.06	7.22 ± 0.06
<R_g_>	12.8 ± 0.2	12.1 ± 0.3	14.2 ± 0.3

**Table 3 biomolecules-10-00197-t003:** Correlations logarithm of the unfolding rate (ln(*ku*)) with protein stability (−(ln*kf* − ln*ku*)) and structural parameters (L, V_asa_/S_asa_, ln(AbsCO) and R_g_) for 108 proteins.

Correlations with ln(*ku*)	All (Two + Multi)	Two-State	Multi-State
Number of proteins	108	70	38
Stability (−(ln*kf* − ln*ku*))	**0.79**	**0.79**	**0.83**
L	−0.71	−0.65	−0.71
V_asa_/S_asa_	**−0.77**	**−0.79**	−0.63
ln(AbsCO)	**−0.84**	**−0.87**	−0.69
R_g_	−0.65	−0.61	−0.60

* Correlations above 0.75 are shown in bold.

**Table 4 biomolecules-10-00197-t004:** Correlations between the logarithm of the unfolding rate (ln(*ku*) and structural parameters (L, V_asa_/S_asa_, ln(AbsCO) and R_g_) for four structural classes of proteins (α, β, α/β and α + β).

Correlations with ln(*ku*)	All (Two + Multi)				Two-State				Multi-State			
	α	β	α/β	α + β	α	β	α/β	α + β	α	β	α/β	α + β
Number of proteins	31	37	12	28	23	25	1	21	8	12	11	7
L	−0.71	**−0.84**	−0.50	−0.60	−0.72	**−0.84**	−	**−0.77**	−0.61	**−0.86**	−0.53	−0.20
V_asa_/S_asa_	**−0.78**	**−0.82**	−0.11	**−0.78**	**−0.84**	**−0.83**	−	**−0.83**	−0.60	−0.66	−0.28	−0.54
ln(AbsCO)	**−0.80**	**−0.89**	−0.30	−0.73	**−0.85**	**−0.90**	−	**−0.83**	−0.48	**−0.77**	−0.29	−0.34
R_g_	−0.73	**−0.83**	−0.69	−0.50	**−0.79**	**−0.84**	−	−0.57	−0.41	−0.74	−0.69	−0.11

* Correlations above 0.75 are shown in bold.

**Table 5 biomolecules-10-00197-t005:** Average values of structural parameters for two-state and multi-state bacterial and eukaryotic proteins.

Average Value	All (Two + Multi)		Two-State		Multi-State	
	Bacteria	Eukaryota	Bacteria	Eukaryota	Bacteria	Eukaryota
Number of proteins	42	53	29	32	13	21
<L>	107 ± 9	92 ± 5	87 ± 10	77 ± 4	152 ± 14	115 ± 10
<V_asa_/S_asa_>	3.37 ± 0.07	3.30 ± 0.04	3.21 ± 0.08	3.20 ± 0.05	3.74 ± 0.07	3.47 ± 0.07
<ln(AbsCO)>	7.10 ± 0.07	7.08 ± 0.06	6.97 ± 0.09	7.04 ± 0.07	7.40 ± 0.07	7.14 ± 0.10
<R_g_>	13.1 ± 0.4	12.9 ± 0.3	12.3 ± 0.5	12.3 ± 0.4	14.8 ± 0.5	13.9 ± 0.4

**Table 6 biomolecules-10-00197-t006:** Correlations between the logarithm of the unfolding rate (ln(*ku*)) and structural parameters (L, V_asa_/S_asa_, ln(AbsCO) and R_g_) for two-state and multi-state bacterial and eukaryotic proteins.

Correlations with ln(*ku*)	All (Two + Multi)		Two-State		Multi-State	
	Bacteria	Eukaryota	Bacteria	Eukaryota	Bacteria	Eukaryota
Number of proteins	42	53	29	32	13	21
L	−0.67	−0.68	−0.61	**−0.75**	−0.43	**−0.77**
V_asa_/S_asa_	−0.72	−0.69	**−0.75**	**−0.77**	−0.11	−0.69
ln(AbsCO)	**−0.80**	**−0.79**	**−0.86**	**−0.77**	−0.22	**−0.81**
R_g_	−0.71	−0.46	−0.64	−0.47	−0.58	−0.50

* Correlations above 0.75 are shown in bold.

**Table 7 biomolecules-10-00197-t007:** Average values of structural parameters for two-state and multi-state thermophilic and mesophilic proteins.

Average Value	All (Two + Multi)		Two-State		Multi-State	
	Thermophile	Mesophile	Thermophile	Mesophile	Thermophile	Mesophile
Number of proteins	10	32	8	21	2	11
<L>	105 ± 18	108 ± 11	82 ± 11	89 ± 13	197 ± 22	144 ± 14
<V_asa_/S_asa_>	3.30 ± 0.14	3.39 ± 0.08	3.13 ± 0.10	3.23 ± 0.10	3.97 ± 0.01	3.70 ± 0.08
<ln(AbsCO)>	7.13 ± 0.15	7.09 ± 0.08	7.03 ± 0.17	6.95 ± 0.11	7.54 ± 0.07	7.38 ± 0.09
<R_g_>	13.3 ± 0.8	13.0 ± 0.5	12.5 ± 0.7	12.2 ± 0.6	16.2 ± 1.1	14.5 ± 0.5

**Table 8 biomolecules-10-00197-t008:** Correlations between the logarithm of the unfolding rate (ln(*ku*)) and structural parameters (L, V_asa_/S_asa_, ln(AbsCO) and R_g_) for two-state and multi-state thermophilic and mesophilic proteins.

Correlations with ln(*ku*)	All (Two + Multi)		Two-State		Multi-State	
	Thermophile	Mesophile	Thermophile	Mesophile	Thermophile	Mesophile
Number of proteins	10	32	8	21	2	11
L	**−0.83**	−0.64	**−0.92**	−0.60	−	−0.27
V_asa_/S_asa_	**−0.77**	**−0.75**	**−0.92**	**−0.78**	−	0.09
ln(AbsCO)	−0.74	**−0.83**	**−0.93**	**−0.85**	−	−0.35
R_g_	**−0.87**	−0.66	**−0.88**	−0.60	−	−0.47

* Correlations above 0.75 are shown in bold.

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
