# Peer review of "How Quickly Do Proteins Fold and Unfold, and What Structural Parameters Correlate with These Values?"

_biomolecules, 2020, doi:10.3390/biom10020197_

Round 1

Reviewer 1 Report

Line 53 – I might use solvent accessible as opposed to just accessible, if this is in fact the case, and define accessible volume more clearly.

Line 54 “relationship between accessible volume to the accessible surface (Vasa/Sasa). The last parameter is proportional to the radius of the smallest section of the protein globule [16]. It was demonstrated that the radius of cross-section (Vasa/Sasa)”. Are radius of the smallest section of the protein globule and radius of cross-section the same?  Terms are unclear.

Why did authors choose to use standard deviation of the mean for  the measurement of the error? Why is error used as all as it is not discussed and not clear why it is relevant.

It may be worthwhile bold or color high correlation numbers in tables for the reader to easily see what is the best correlation (e.g. values above 0.80 or 0.75 or 0.70  to highlight your conclusions)

Figure 7.  What does “observed characteristic rate of approximation of the equilibrium between the native and the unfolded structures”. What is a rate of equilibrium?  Is this a rate constant or an equilibrium constant?

Why is color used only at end of manuscript?  Color should be used consistently.

A little more discussion on significance of results is needed.

Author Response

Reviewer 1: We are grateful for the constructive comments that arose during the review process.

Line 53 – I might use solvent accessible as opposed to just accessible, if this is in fact the case, and define accessible volume more clearly.

Answer: We have added the word «solvent» to the accessible surface and accessible volume. We also have clarified the definition of the solvent accessible volume. 

Line 54 “relationship between accessible volume to the accessible surface (Vasa/Sasa). The last parameter is proportional to the radius of the smallest section of the protein globule [16]. It was demonstrated that the radius of cross-section (Vasa/Sasa)”. Are radius of the smallest section of the protein globule and radius of cross-section the same?  Terms are unclear.

Answer: The radius of the smallest section of the protein globule and radius of cross-section are not the same. We have clarified this in the text.

Why did authors choose to use standard deviation of the mean for  the measurement of the error? Why is error used as all as it is not discussed and not clear why it is relevant.

Answer: Errors were calculated as in the early papers PloS One, 2009, 4, e6476; PLoS One, 2011, 6(12), e28464.

It may be worthwhile bold or color high correlation numbers in tables for the reader to easily see what is the best correlation (e.g. values above 0.80 or 0.75 or 0.70  to highlight your conclusions)

Answer: In Tables the correlations above 0.75 have been bolded.

Figure 7.  What does “observed characteristic rate of approximation of the equilibrium between the native and the unfolded structures”. What is a rate of equilibrium?  Is this a rate constant or an equilibrium constant?

Answer: We have clarified the definition. This is the constant of observed rate (kapp=kf+ku).

Why is color used only at end of manuscript?  Color should be used consistently.

Answer: We have added color to the all figures of the paper.

A little more discussion on significance of results is needed.

Answer: We have added a little more discussion on significance of the results.

Reviewer 2 Report

The authors present a study to determine if connection exist between protein unfolding rates and structural parameters. This manuscript has been significantly improved and some points have been clarified in the revised document. The manuscript remains in need of significant revisions. This reviewer recommends publication after major revisions.

Here are my general comments.

Avoid heavy usage of 1st person. The materials and methods needs more information as results cannot be replicated without it. For instance, which databases were these parameters taken from? Which proteins and what was the decision making process in determining which proteins to add to the study? What software programs performed the data analyses.

Comments from the text.

Line 36, "in the papers" should be changed and the whole sentence reworded to make sense. Lines 60-73 need to be reworded to flow better in the introduction. The text seems choppy. Also, please define "golden triangle." Lines 120-122. Rework this sentence so that multi-state values go first and then 2-state values since the sentence references multi-state first. Line 254, a typo. "termophilic"

Author Response

Reviewer 2: We thank you very much for the comments and suggestions. The comments and suggestions are valuable and very helpful for revising and improving our manuscript.

The authors present a study to determine if connection exist between protein unfolding rates and structural parameters. This manuscript has been significantly improved and some points have been clarified in the revised document. The manuscript remains in need of significant revisions. This reviewer recommends publication after major revisions.

Here are my general comments.

Avoid heavy usage of 1st person. The materials and methods needs more information as results cannot be replicated without it. For instance, which databases were these parameters taken from? Which proteins and what was the decision making process in determining which proteins to add to the study? What software programs performed the data analyses.

Answer: Experimentally measured in vitro folding and unfolding rate constants in water were taken in consideration. A database of such proteins began to be collected since 2003. At that time there were only 57 proteins. In 2009, there were already 84 proteins. Now this database consists of 108 proteins. We have added all this information to the text. In Supplementary Table S1 contains all data (folding and unfolding rates and structural parameters of the proteins: L, Vasa/Sasa, ln(AbsCO) and Rg) for reproducing the results obtained in the paper. We have added equations for calculating of ln(AbsCO) and Rg. Parameters Vasa and Sasa were calculated using the YASARA program.

Comments from the text.

Line 36, "in the papers" should be changed and the whole sentence reworded to make sense. Lines 60-73 need to be reworded to flow better in the introduction. The text seems choppy. Also, please define "golden triangle." Lines 120-122. Rework this sentence so that multi-state values go first and then 2-state values since the sentence references multi-state first. Line 254, a typo. "termophilic"

Answer: We have reworded these sentences in the text.

Round 2

Reviewer 2 Report

The authors have submitted a revised document with details of the proteins under investigation which was major deficit in a previous submission. With the additional information added, the manuscript is in good shape for publication. This reviewer recommends publication.

This manuscript is a resubmission of an earlier submission. The following is a list of the peer review reports and author responses from that submission.

Round 1

Reviewer 1 Report

The author present an argument that shows how folding and unfolding rates of proteins correlate with structural parameters. While this study certainly has scientific merit, this reviewer recommends rejection of the manuscript for the following reasons:

The structural parameters that are not clearly defined making it difficult for the reader to follow the significance of the findings. The introduction needs to be entirely revised to expand on structural parameters and other studies that are referenced in the introduction but not clearly explained. Each paragraph needs to have more explanation of the literature and how it relates to the current study. The materials and methods are not sufficient for a researcher to replicate the experiments described. The only information given is the database and the number of each type of protein, which is clearly not enough information for replication. The conclusion should wrap up the study in the form of a table or some other visual that organizes the results in way that makes it easily accessible to other researchers in the field.

Reviewer 2 Report

The manuscript is very descriptive but does not put the results in context.  How well do unfolding rates correlate with folding rates and were the correlations of folding rates with structural factors similar or different.  Does the structure of the protein in terms of alpha and alpha/beta make a difference in unfolding rates as it does for folding rates?  Which of these structural factors was most important for folding rates and are they similar for unfolding rates.

The paper was also more challenging than most to read for a variety of reasons.

Website with references to where the data was taken from was not accessible. Need to add references to excel spreadsheet, as some of the numbers varied from previously reported data but may represent an update.

Grammar and clarity could be improved such as one and two sentence paragraphs. ALL Abbreviations need to be defined when initially used not long afterwards or at all, e.g L, Vasa/Sasa, Rg, ku, kf or kmt etc.  What is kmt ?

What do the errors throughout the manuscript correspond to especially in Tables 1, 3 etc.? Standard deviations? Results in Table 3 suggest something else.

Title of Table 2 contains folding rates but not clear how the data is incorporated into table?

Figures 2, 4 etc. should have best fit line and equation or statement about why they are not there.

Table 5. How do you have a standard deviation with only two points (thermophile multistate).  If several data points were used for same protein, then why is correlation coefficient 1.

Correlation between folding and unfolding rates would suggest that protein stability is similar or varies in linearly fashion with folding rates. Is this the case?

Stability also needs to be addressed more clearly for thermophilic versus mesophilic. What is kapp in Chevron plot?

Much of the data in the Tables is repeated in the text.